# 3,5-DCQA as a Major Molecule in MeJA-Treated *Dendropanax morbifera* Adventitious Root to Promote Anti-Lung Cancer and Anti-Inflammatory Activities

**DOI:** 10.3390/biom14060705

**Published:** 2024-06-15

**Authors:** Fengjiao Xu, Anjali Kariyarath Valappil, Shaojian Zheng, Bingsong Zheng, Deokchun Yang, Qiang Wang

**Affiliations:** 1State Key Laboratory of Plant Environmental Resilience, College of Life Sciences, Zhejiang University, Hangzhou 310058, China; xufengjiao2020@163.com (F.X.); sjzheng@zju.edu.cn (S.Z.); 2School of Biological and Chemical Engineering, NingboTech University, Ningbo 315100, China; 3Department of Biopharmaceutical Biotechnology, College of Life Science, Kyung Hee University, Yongin-si 17104, Republic of Korea; x17860771929@163.com; 4State Key Laboratory of Subtropical Silviculture, Zhejiang A&F University, Hangzhou 311300, China; bszheng@zafu.edu.cn

**Keywords:** *D. morbifera*, adventitious root, methyl jasmonate, 3,5-Di-caffeoylquinic acid, anti-inflammation, anti-lung cancer

## Abstract

(1) Background: Phytochemicals are crucial antioxidants that play a significant role in preventing cancer. (2) Methods: We explored the use of methyl jasmonate (MeJA) in the in vitro cultivation of *D. morbifera* adventitious roots (DMAR) and evaluated its impact on secondary metabolite production in DMAR, optimizing concentration and exposure time for cost-effectiveness. We also assessed its anti-inflammatory and anti-lung cancer activities and related gene expression levels. (3) Results: MeJA treatment significantly increased the production of the phenolic compound 3,5-Di-caffeoylquinic acid (3,5-DCQA). The maximum 3,5-DCQA production was achieved with a MeJA treatment at 40 µM for 36 h. MeJA-DMARE displayed exceptional anti-inflammatory activity by inhibiting the production of nitric oxide (NO) and reactive oxygen species (ROS) in LPS-induced RAW 264.7 cells. Moreover, it downregulated the mRNA expression of key inflammation-related cytokines. Additionally, MeJA-DMARE exhibited anti-lung cancer activity by promoting ROS production in A549 lung cancer cells and inhibiting its migration. It also modulated apoptosis in lung cancer cells via the Bcl-2 and p38 MAPK pathways. (4) Conclusions: MeJA-treated DMARE with increased 3,5-DCQA production holds significant promise as a sustainable and novel material for pharmaceutical applications thanks to its potent antioxidant, anti-inflammatory, and anti-lung cancer properties.

## 1. Introduction

Inflammation serves as the body’s protective response to environmental threats of infection. Additionally, it plays a role in developing and advancing various human diseases, including cancer [1]. Epidemiologic research has shown that inflammation-induced tissue damage may play a role in the initiation and progression of lung cancer, especially when combined with tobacco use. Studies indicate a higher risk of lung cancer in individuals with lung infections like tuberculosis, bacterial pneumonia, or inflammatory lung diseases [2]. Additionally, elevated levels of C-reactive protein, a marker of inflammation, are linked to an increased likelihood of developing lung cancer. In essence, inflammation appears to contribute to the development and promotion of lung cancer, particularly in conjunction with smoking [3]. Lung cancer is the second most diagnosed cancer in both men and women, but it remains the leading cause of cancer-related deaths. In 2020, it claimed more lives than breast, colorectal, and prostate cancers combined. According to a study by Khosravifarsani et al. (2023), the World Health Organization reported that lung cancer resulted in 1.8 million fatalities and saw 2.21 million new cases worldwide. Presently, the treatment options for lung cancer are not only expensive but also ineffective due to the significant side effects and toxicity they impose on healthy tissues. As a result, there is a pressing need to investigate a more cost-effective and biocompatible therapeutic approach for combating lung cancer [4]. Currently, multiple strategies for neoplasm treatment are employed in clinical settings, including surgery, radiotherapy, chemotherapy, immunotherapy, and laser therapy. Despite these advancements, chemotherapy remains a key option for cancer treatment. The primary goal of chemotherapy is to selectively kill malignant cells without causing significant harm to normal cells; however, some healthy cells are also destroyed, leading to substantial side effects such as bone marrow suppression, hair loss, and gastrointestinal issues [5]. Many clinically significant nucleoside/non-nucleoside-related analog drugs, such as gemcitabine, 5-fluorouracil, doxorubicin, tamoxifen, etc., have been widely used and have played a crucial role in cancer chemotherapy. However, these drugs have notable side effects, such as anthracyclines, which can cause cardiac toxicity in patients [6]. Recent evidence indicates that drug resistance accounts for 90% of chemotherapy failures and remains a major limiting factor in curing cancer patients. Cancer chemoprevention offers a hopeful strategy for decreasing both the occurrence and fatality rates of lung cancer. This approach involves utilizing dietary, natural, or pharmaceutical substances to hinder cancer initiation, mitigate DNA damage, impede the advancement of tumors in precancerous cells, and reduce metastasis risk, all while minimizing adverse effects [7]. Recently, there has been a transition towards investigating new plant-based phytochemicals, whether they are nutritive or non-nutritive, possessing antioxidant and anticancer properties for inflammation and cancer treatment [8].

*D. morbifera*, a broad-leaved evergreen tree, belonging to the Araliaceae family, can be found in the southwestern part of East Asia, Republic of Korea, and Japan [9,10,11]. Different parts (such as seeds, leaves, roots, and stems) of this plant species have been used in traditional medicine for treating headaches, infectious disorders, skin problems, and other afflictions [12]. The genus *Dendropanax* has been reported to contain polyphenols, essential oils, phenolics, flavonoids, tannins, terpenoids, and alkaloids [13]. Previous studies have identified a variety of biological activities of *D. morbifera*, such as antioxidant [14], anti-inflammatory [15], anti-amnestic [16], and neuroprotective activities [17]; anticancer potential for various tumor cell lines, including colon adenocarcinoma cells (COLO-205), clonal human osteosarcoma cells (HOS), biliary tract cells (SNU-245 and SNU-308), and hepatocellular carcinoma cells (Huh-BAT and Huh-7) [18]; and anti-diabetic [19,20], hepatoprotective, immunomodulatory [21], antibacterial, antifungal [22], anti-plasmodial [23], cytotoxic, and larvicidal functions [24].

From the points mentioned above, research on *D. morbifera* needs to be carried out continuously, and it is suited to work as a natural component for developing novel pharmaceuticals. Therefore, ongoing cultivation and production are required. *Dendropanax* seeds, however, have a low tolerance for cold stress. It takes over six years of flowering and fruit production before seeds can be harvested [25]. Additionally, it has been noted that seeds obtained by a protracted method show relatively poor germination rates [26]. Due to the geographical restrictions of *D. morbifera*, it is not easy to cultivate in communities in other areas, and it takes more time for seed production [27]. There is limited research on the in vitro propagation of *Dendropanax* species [28,29], especially about the compound obtained from in vitro propagation materials.

There have been a lot of studies on how to enhance secondary metabolite accumulation using various biotechnologies. Elicitors, precursors, bioinformation, environmental stresses, and changes in medium ingredients are used to increase the yield of secondary metabolites [30]. Jasmonic acid (JA) and its derivative methyl jasmonate (MeJA) are plant hormones that have been widely used for induction studies in in vitro culture systems. The priming process leads to crosstalk between JA and its receptors in the plasma membrane, which results in a series of cellular defense responses and the induction of oxidative stress-protective enzymes [31]. Additionally, this also leads the cell to produce and accumulate signaling molecules, which in turn regulate the expression of genes involved in the formation of secondary metabolites; therefore, these signaling molecules have been employed in in vitro elicitation investigations [32,33,34]. However, MeJA is the most commonly used inducer, demonstrating a notable effect of accumulating secondary metabolites in plant cell and organ cultures [31,33].

In this study, we cultured *D. morbifera* in vitro for the biomass of adventitious roots under optimal conditions, the compounds of which were extracted and subsequently compared with the leaf extracts. What is more, a MeJA elicitor was applied for the enhancement of certain phenolic compounds. The anti-inflammatory activity was checked on RAW 264.7 cell lines and anti-lung cancer activity on A549 cell lines. Both pharmacological activities were enhanced after the treatment with MeJA. Remarkably, as far as our current knowledge extends, this marks the first documented use of the MeJA elicitor in the adventitious root culture of *D. morbifera* to enhance the production of secondary metabolites. This breakthrough widens the range of potential applications for this plant in pharmaceutical research, enabling more efficient and rapid utilization.

## 2. Materials and Methods

### 2.1. Materials and Chemicals

The leaves of five-year-old *D. morbifera* were obtained from Jangseong-gun, Jeollanam-do (35°17′52″ N 126°47′04″ E), Republic of Korea. McCown Woody Plant Medium (WPM), plant agar, sucrose, 2,4-Dichlorophenoxyacetic acid (2,4-D), N6- benzyl adenine (BA), and Indole-3-butric acid (IBA) were provided by Duchefa Biochemie company (Haarlem, The Netherlands). Elicitors including methyl jasmonate (MeJA), chitosan (CHI), yeast extract (YE), and salicylic acid (SA) were supplied by Merk. DPPH (2,2-diphenyl-1-picryl-hydrazyl), potassium ferricyanide, trichloroacetic acid, and ferric chloride were purchased from Sigma-Aldrich (Saint Louis, MO, USA). 3,5-DCQA was purchased from Chengdu Refmedic Biotechnology Co., Ltd., Chengdu, China. The murine macrophage (RAW 264.7) and lung cancer cell lines (A549) were provided by the Korea Cell Line Bank (KCLB, Seoul, Republic of Korea). Dulbecco’s modified Eagle medium (DMEM) containing L-glutamine, RPMI 1640 culture medium (with L-glutamine), and PBS (phosphate-buffered saline, pH 7.4) were purchased from Welgene Inc. (Gyeongsan-si, Republic of Korea). Penicillin/streptomycin solution and fetal bovine serum were obtained from GenDEPOT. 3-(4,5-dimethylthiazol-2-2yl)-2,5-diphenyl tetrazolium bromide (MTT) was purchased from Gibco (Waltham, MA, USA).

### 2.2. Tissue Culture of D. morbifera

The procedure of mass induction of *D. morbifera* adventitious roots (DMAR) followed that of a previous study [29] with little modifications. Briefly, tap water was used to remove the dust, and then the leaves were surface sterilized using 70% ethanol (*v*/*v*) for 1 min, washed with 2% (*v*/*v*) sodium hypochlorite solution for 15 min, and then washed with distilled water 3 times. After sterilization, the leaf tissues were cut into 0.5 cm pieces for each, wounded, and plated to solid WPM containing 30 g/L sucrose, 2,4-D 0.5 mg/L, BA 2 mg/L, and 7 g/L plant agar. The pH of the medium was adjusted to 5.7, sterilized at 121 °C for 20 min, and then dispensed into a Petri dish (90 mm × 20 mm). The cultivation state was maintained at room temperature (23 ± 1 °C) and 40% humidity indoors and cultured in a dark room where the light was completely blocked. When callus was successfully induced, we then transferred the callus to solid WPM containing 30 g/L sucrose, IBA 3 mg/L, and 7 g/L plant agar with the same cultivation conditions as mentioned above. When the adventitious roots were obtained, 1 g of adventitious roots was inoculated into a flask containing 100 mL WPM supplemented with 30 g/L sucrose and IBA 3 mg/L. The inoculated Erlenmeyer flask was incubated in a dark room at 110 rpm for 4 weeks in the shaking incubator (1 rpm = 1/60 Hz).

### 2.3. Preparation of Leaf and DMAR Extracts (DMLE and DMARE)

After harvesting the DMAR, with leaves, they were dried and ground into powder. According to previous research, 70% EtOH was used for extracting this plant [35]. We chose 70% EtOH for extraction using a sonication method under 80 °C for 1 h. Then, the extracts were centrifuged at 7000 rpm for 10 min, and, subsequently, the supernatant was collected after fliting by Whatman filter paper no. 2. Each group was repeated 3 times. The mixtures obtained from the 70% EtOH extraction method were concentrated using a rotary evaporator, and, after removing all of the solvent, the powder was collected and kept at 4 °C for further use.

### 2.4. Compounds Analysis by HPLC and LC-MS

The powder, collected in the previous step as described in the preceding section, was weighed, dissolved in distilled water, and then filtered through a 0.45 μm syringe filter to achieve a solution with the same concentration. This filtered solution was subsequently loaded into the high-performance liquid chromatography (HPLC) equipment and the liquid chromatography-mass spectrometry (LC-MS) equipment. The analysis was conducted using an Agilent 1260 Infinity system, which included components such as a quaternary pump (G1311B), standard autosampler (G1329B), column thermostat compartment (G1316A), variable wavelength detector (G1314F), and a ZORBAX Eclipse Plus C18 column (250 mm × 4.6 mm, 5 µm particle size) from Milford, MA, USA. The column temperature was maintained at 35 °C. The mobile phase was solvent A (0.1% acetic acid) and solvent B (Methanol). 

For HPLC analysis, the mobile phase B was programmed as follows: 0–8 min, 90–80%; 8–30 min, 80–55%; and 30–60 min, 55–30%, at a flow rate of 1 mL/min with an injection volume of 5 μL. In the LC-MS analysis, the mobile phase A was set as follows: 0–8 min, 80%; 8–30 min, 55%; 30–60 min, 30%; 60–60.5 min, 90%; and 60.5–65, 90%, at a flow rate of 0.3 mL/min with an injection volume of 5 μL. An electrospray ionization (ESI) negative model was used to measure the chemicals extracted from the *D. morbifera* materials.

### 2.5. Elicitors on DMAR for Enhancing Secondary Metabolites

To investigate the effects of different elicitors such as MeJA, SA, CHI, and YE on the accumulation of secondary metabolites, various concentrations of each elicitor were applied to the DMAR with an initial inoculum of 5 g. The DMAR was treated by MeJA at a concentration of 100 μM and SA, CHI, and YE at 100 mg/L. The effects were determined by checking the biomass of DMAR, and antioxidant activity was determined through DPPH and reducing power after a 7-day culture. 

After the confirmation of the increased compound by HPLC and LC-MS, a time course assay (0–48 h) and the determination of the effect of the concentration of MeJA (0–100 μM) on the enhancement of the increased compound were carried out by dipping 5 g DMAR into 100 mL WPM supplemented with MeJA. Each treatment group was set to 3 replications. After harvesting, the DMAR extracts were prepared to evaluate the increased compound by using HPLC and LC-MS. After the confirmation of the compound that was enhanced by the treatment of the elicitor, the amount of this compound was determined using the calibration curve (in the concentration range of 0.0625–1 mg/mL and R^2^ = 0.9922 (n = 3)). The linear regression of absorption was Y=5906.7X−187.9.

### 2.6. Total Phenolic Content (TPC) Measurement

The previous methodology for measuring the TPC was applied with little modification [36]. Briefly, 30 μL of sample and 150 μL of 10% 2 N Folin–Ciocalteu reagent were mixed. The mixture was vortexed thoroughly and kept for 5 min, and, after that, 160 μL of 7.5% sodium carbonate solution was added. The mixture was subsequently kept in the dark for 60 min. Then, 100 μL of the mixture was taken and added into corresponding wells of a 96-well microplate with 3 replications. The absorbance was read at 715 nm. The TPC was calculated from a standard curve by using gallic acid as the standard. The results were given as μg of Gallic acid equivalent (GAE) per mg of extract.

### 2.7. Total Flavonoid Content (TFC) Measurement

The TFC was detected based on the aluminum chloride colorimetric method reported previously with slight modifications [37]. Briefly, 50 μL of the sample was mixed with 430 μL of distilled water, 10 μL of 1 M potassium acetate, and 10 μL of 10% aluminum chloride. The mixture was then extensively vortexed. The mixture was incubated at room temperature for 30 min after spinning down. Then, 100 μL of the mixture was taken and added into corresponding wells of a 96-well microplate with 3 replications. The absorbance was read at 415 nm. The TFC was calculated from a standard curve by using rutin as standard. The results were given as μg of Rutin equivalent (RE) per mg of extract.

### 2.8. Antioxidant Activity

#### 2.8.1. DPPH

The previous methodology for measuring the antioxidant activity by DPPH was applied with minor modifications [36]. Briefly, 30 μL of the sample was mixed with 270 μL of 0.2 mM DPPH and incubated at room temperature in the dark for 30 min. Then, 100 μL of the mixture was taken and added into corresponding wells of a 96-well microplate with 3 replications. The absorbance was read at 517 nm. The percentage inhibition of the sample was calculated from a standard curve by using gallic acid as the standard. The calculation was carried out as follows: Inhibition (%)=[((Acontrol−Asample))/Acontrol]×100.

#### 2.8.2. Reducing Power

The antioxidant activity of the sample can also be determined by checking the reductant ability of plant extracts [37]. In this method, a mixture of 100 μL of the sample, 250 μL of phosphate buffer (pH 6.6), and 250 μL of potassium ferricyanide was applied. The mixture was vortexed thoroughly and incubated at 50 °C for 20 min, and, after that, 250 µL of 10% trichloroacetic acid was added to the mixture and then centrifuged at 3000 rpm for 10 min. Subsequently, a mixture of 50 µL of the supernatant, 10 µL of 0.1% ferric chloride solution, and 50 µL of distilled water was added into corresponding wells of a 96-well microplate with 3 replications. The absorbance was read at 517 nm. The results were given as μg of gallic acid equivalent (GAE) per mg of extract.

### 2.9. Cell Cultures

A human lung carcinoma cell line (A549) was cultured in 89% RPMI 1640 medium (with L-glutamine), and murine macrophage (RAW 264.7) cells were cultured in a L-glutamine DMEM medium. Both cell lines were cultured in a humidified incubator at 37 °C with a 5% CO_2_ atmosphere in a specific medium supplemented with 10% FBS and 1% penicillin/streptomycin and allowed to adhere and develop for 24 h before treatment with different samples.

### 2.10. Cytotoxicity Assay

The cytotoxicity of DMARE, MeJA-DMARE, and 3,5-DCQA was measured using a 3-(4,5-dimethylthiazol-2-2yl)-2,5-diphenyltetrazolium bromide (MTT) colorimetric assay in RAW 264.7 and A549 cells. Both cell lines were placed and cultured for 24 h in a 96-well plate at a density of 2 × 10^4^ cells per well at 37 °C in a humidified environment with 5% CO_2_. Cells were then added to various concentrations (62.5, 125, 250, 500, and 1000 μg/mL) of DMARE and MeJA-DMARE and incubated for 24 h. After 24 h of treatment, the supernatant was discarded, and the cells were washed with PBS 3 times carefully. After that, 20 μL of MTT (5 mg/mL in PBS) was added to the cells and incubated at 37 °C or 3 h. The supernatant was removed, and 100 μL of DMSO was used in each well to dissolve the in-soluble formazan in a dark condition. The DMSO solution was used as a blank value to be subtracted from experimental values. The cell viability percentages were calculated using the following formula: $Cell\;viability\%=\left(\frac{A\;sample}{A\;control}\right)\times 100$. The optical density values were read by a microplate reader (Bio-Tek, Instruments, Inc., Winooski, VT, USA) at 570 nm. 

### 2.11. Inhibition of NO Production 

The nitric oxide (NO) inhibition assay was conducted with a minor modification compared to the previous method [38]. RAW 264.7 cells were pretreated for 1 h with different concentrations of DMARE, MeJA-DMARE, and 3,5-DCQA. After that, cells were stimulated with 1 μg/mL Lipopolysaccharide (LPS) before being incubated for 24 h. Grise’s reagent was applied to measure the amount of nitrite present in the medium. Briefly, 100 μL of supernatant and 100 μL of Griess reagent were mixed. Then, absorbance was determined at 540 nm using a microplate reader (BioTek Instruments, Inc., Winooski, VT, USA). L-NG-monomethylarginine acetate salt (L-NMMA) was utilized in this experiment at a dose of 50 μM as a positive control (standard inhibitor). The data were represented as NO production (%) after three runs of each experiment.

### 2.12. Reactive Oxygen Species (ROS) Generation Assay

Reactive oxygen species (ROS) intensity on A549 cells was measured using 2′,7′-dichloro-dihydro-fluorescein diacetate (DCFH-DA) [39]. A594 cells (1 × 104 cells/well) were seeded on a 96-well plate and incubated at 37 °C with 5% CO_2_ for 24 h. Following seeding, the cells were exposed to 125 µg/mL and 250 μg/mL of DMARE, MeJA-DMARE, and 3,5-DCQA for ROS generation. Conversely, RAW 264.7 cells were exposed to the samples following a 1 μg/mL LPS-induced inflammation. Following a 24-h treatment period in a serum-free medium, the cells were exposed to 20 µM DCFH-DA at 37 °C under dark conditions for 30 min. After discarding the old medium, 100 mL of PBS was used to wash the cells twice. Using a multi-model plate reader (fluorescence spectrometer), the fluorescence intensity of ROS production at an excitation wavelength of 485 nm and an emission wavelength of 528 nm was calculated. The DCFH-DA reagent was used to detect the rise in ROS.

### 2.13. Quantitative Reverse Transcription-Polymerase Chain Reaction (qRT-PCR)

Total RNA was isolated using a QIAzol lysis reagent (QIAGEN, Germantown, MD, USA) from RAW 264.7 and A549 cell lines. For cDNA synthesis, 1 µg of isolated total RNA was added to a 20 µL reaction mixture using the amfifiRivert reverse transcription kit (GenDepot, Barker, TX, USA) according to the manufacturer’s instructions. The process of cDNA synthesis was carried out under the following conditions: 25 °C for 5 min, 42 °C for 59 min, and 70 °C for 15 min. SYBR TOPreal qPCR 2X Premix (Enzynomics, Daejeon, Republic of Korea) was added to perform qRT-PCR. Briefly, the reactions were carried out in triplicate and required a final volume of 10 L with 2× Master Mix, 1 µL of template cDNA, and 1 µL of forward and reverse primers. CFX Connect Real-Time PCR (Bio Rad, Hercules, CA, USA) was used for all real-time measurements. The following conditions were used for amplification reactions: 95 °C for 10 min, then 40 cycles of 95 °C for 20 s and 55 °C for 30 s, and then 72 °C for 15 s. The comparative 2^−ΔΔCt^ method was used to measure the relative amounts of mRNAs, and the GAPDH gene was used to standardize the results.

### 2.14. Wound Healing Assay

We performed a wound-healing assay to assess the migratory potential of A549 cancer cells. In brief, A549 lung cancer cells were seeded into 6-well plates at a density of 2 × 10^4^ cells per well and incubated at 37 °C for 1 day. Subsequently, a vertical scratch was created in the cell monolayer using a 10 µL sterile pipette tip, and detached cells were removed with PBS. The cells were then exposed to DMAR, MeJA-AMAR, and 3,5-DCQA at 250 µg/mL. After a treatment period of 1 day, photographs were captured using an implanted 5.0-megapixel MC 170 HD camera from Wetzlar, Germany.

### 2.15. Statistical Analysis

We employed GraphPad 9.0 software for data analysis (GraphPad Software, San Diego, CA, USA). In the case of TPC, TFC, DPPH, and reducing power assays, we determined the significance of differences between samples using the Tukey (HSD) test at a significance level of 0.05. For in vitro experiments, mean values were subjected to a two-way analysis of variance (ANOVA) with Dunnett’s test for comparisons. It is important to note that all experiments were independently repeated a minimum of three times unless otherwise indicated. Statistical significance was denoted as follows: * for *p* < 0.05, ** for *p* < 0.01, and *** for *p* < 0.001, as well as # for *p* < 0.05, ## for *p* < 0.01, and ### for *p* < 0.001.

## 3. Results and Discussions

### 3.1. Adventitious Roots Induction and Antioxidant Activity Evaluation 

After a 4-month culture, the biomass production of DMAR was obtained, as shown in Figure 1. The basal medium of woody plants usually contains smaller amounts of macronutrient salts, and 1/2 MS, MMS (modified MS), and WPM are now widely used in woody plants. The 1/2 MS or MMS media do not seem to cause ammonium toxicity when only half of the MS ammonium is present. Like MS, WPM contains less total nitrogen and ammonium [40]. 

The TPC, TFC, and antioxidant activities of DMARE and DMLE were compared. As shown in Table 1, the TPC of DMLE (140.2 ± 1.9 µg GAE/mg extracts) was higher than that of DMARE (116.5 ± 1.4 µg GAE/mg extracts). Conversely, DMARE exhibited a higher TFC (105.8 ± 12.4 µg RE/mg extracts) compared to DMLE (98.3 ± 2.6 µg RE/mg extracts). Both DMLE and DMARE demonstrated similar antioxidant activity, with DPPH results showing values of 8.7 ± 0.1 µg GAE/mg extracts for DMARE and 8.5 ± 0.4 µg GAE/mg extracts for DMLE. Similarly, the reducing power assay indicated that DMLE possessed superior antioxidant activity (315.7 ± 7.0 µg GAE/mg extracts), albeit only slightly higher than that of DMARE (311.5 ± 26.3 µg GAE/mg extracts). Therefore, DMARE derived from in vitro culture is a valuable supplement for pharmaceutical materials, particularly from an antioxidant perspective.

### 3.2. Effects of Elicitors on the Compound Enhancement of DMARE

#### 3.2.1. Chemical Composition

To gain deep insights into why our DMARE exhibited good antioxidant activities, we conducted HPLC to identify the compositions of DMLE and DMARE. Figure 2A–C shows that five compounds (**1**–**5**) were detected in DMLE. Among these, compounds **1**, **2**, **3**, and **5** were identified as Neo-chlorogenic acid, Chlorogenic acid, Crypto-chlorogenic acid, and Rutin, respectively. On the other hand, four compounds were observed in DMARE, with compound **2** being chlorogenic acid and compound **4** also present. Notably, compounds **6** and **7** were exclusively found in DMARE. 

#### 3.2.2. The Effect of Elicitors

When exposed to in vitro stress factors known as elicitors, which are signals that cause the synthesis of secondary metabolites, it has been discovered that plants produce a variety of defensive responses [41]. In previous studies, MeJA, SA, CHI, and YE showed positive comebacks to the synthesis and/or enhancement of phenolic compounds in various plant species [42,43,44,45]. Various elicitors such as MeJA, SA, CHI, and YE were employed to enhance the production of secondary metabolites. However, according to our results, only MeJA showed a positive effect in increasing phenolic compound accumulation; there was almost no difference after the treatment of SA, CHI, and YE. As illustrated in Figure 2D, one compound increased with the treatment of MeJA at 100 μM. Subsequently, LC-MS was applied to identify the compound in MeJA-DMARE, and, according to the negative mode of LC-MS reports, we deduced that the increased compound **4** was 3,5-Di-caffeoylquinic acid (3,5-DCQA) (Figure 2F). To support our research, a similar function of MeJA on the enhancement of 3,5-DCQA was demonstrated in one study by Wei et al. [46]. The compounds presented only in DMARE were both Di-caffeoylquinic acid isomers.

The growth and biomass were assessed by measuring fresh weight, dry weight, and the amount of 3,5-DCQA after treatment with MeJA. As illustrated in Figure 3A–C, when 5 g of DMAR was inoculated in liquid WPM supplemented with these MeJA after 7 days of culture, the biomass was largely inhibited (fresh and dry weight were 6.1 ± 0.9 g and 0.40 ± 0.1 g, respectively); however, the 3,5-DCQA was increased (9.0 ± 0.10 (μg/mg extract)), which was 1.7 times as high as that of the untreated DMARE. Similar results were obtained in previous studies, in which it was discovered that MeJA inhibits the accumulation of biomass in cell suspension cultures of Taxus [47] and strawberries [48]. MeJA suppresses the mitotic cycle in plant cells, stopping them in the G1 phase before the transition to the S-phase, as demonstrated by two separate investigations. This slows the progression of the cell cycle, which lowers the quantity of actively dividing cells [47,49]. 

In response to various elicitor treatments, secondary metabolite content in the adventitious root suspension culture was estimated using the biochemical markers DPPH and reducing power. The antioxidant activities were quantified. As indicated in Figure 3D,E, the MeJA treatment group exhibited higher antioxidant activity than DMARE no matter the DPPH results (9.3 ± 0.3 μg GAE/mg extracts) or reducing power results (185.9 ± 4.6 μg GAE/mg extracts). It has already been established and documented that MeJA affects plants’ secondary metabolism [50]. According to [51,52], phenolics and flavonoids are often antioxidative components formed via the phenylpropanoid route. 

Samaneh Attaran Dowom et al. studied the effect of MeJA on the content of TPC and TFC in regenerated shoot cultures of *Salvia virgata* Jacq. The results indicated that MeJA exhibited a significant capacity to increase the formation of important phenolic acids [53]. In another study, Abinaya Manivannan et al. investigated the effect of different elicitors, such as MeJA, SA, and sodium nitroprusside (SNP), on the buildup of secondary metabolites and antioxidant abilities. The results showed that MeJA was one of the elicitors that increased the levels of total phenolics and flavonoids [54].

#### 3.2.3. Effects of Concentration and Elicitation Time of MeJA on 3,5-DCQA Production

Further, the concentration and work time of MeJA were evaluated to choose the optimal condition for the enhancement of compound 3,5-DCQA. As shown in Figure 3F, when the DMAR was treated with 40 μM MeJA for 1 day, the highest amount of 3,5-DCQA could be obtained, and the amount was 10.927 ± 0.260 μg/mg extract, which was around 1.756 times as high as that of the untreated group. MeJA had a dose-dependent effect on the accumulation of 3,5-DCQA, and treatment with different concentrations of MeJA from 20 μM to 100 μM all showed a positive influence. However, when the concentration of MeJA was over 40 μM, the favorable effect of MeJA was diminished; especially at 100 M, the amount of 3,5-DCQA was only 1.14 times as high as that of the control.

As for the time course assay, when the DMAR was treated with MeJA at a concentration of 40 μM for 36 h, the maximum amount of 3,5-DCQA could be attained, which was 2.111 times higher than that of the untreated group (Figure 3G). MeJA elicitation treatment also exhibited a time-dependent response on the accumulation of 3,5-DCQA. When the DMAR was exposed to MeJA for various periods, the generation of 3,5-DCQA increased with time. When the elicitation time with MeJA reached 24 h, 3, 5-DCQA increased slowly.

Secondary metabolite production is significantly influenced by elicitor concentration and exposure duration [49]. Jasmonates often generate eustress at low concentrations, stimulating transcriptional cascades and intracellular manufacturing of protective chemicals, while at high concentrations (50–300 M), they boost the growth of distress, senescence, and cell death in plants [55]. In addition, MeJA is generated in stress response, which activates defense and programmed cell death pathways in plants. Zhang et al. investigated the signaling pathways that cause cell death when MeJA is applied. They measured stress events and energy utilization efficiencies using chlorophyll-delayed fluorescence. They found that delayed fluorescence significantly decreased after each MeJA treatment, particularly after 36 h, with reductions becoming more pronounced at higher concentrations. Furthermore, MeJA treatments resulted in severe decreases in plant development after 72 or 108 h, although the impact on growth was smaller than the observed decreases in delayed fluorescence intensity [55]. In this case, this may be one of the reasons why there was little difference in the induced 3,5-DCQA after 36 h of MeJA treatment.

### 3.3. Cytotoxicity Effect of DMARE and MeJA-DMARE

The cytotoxicity test, one of the biological evaluation and screening procedures, employs tissue cells in vitro to assess the effects of medical materials on cell growth, reproduction, and morphology. Cytotoxicity is a vital technique in biological evaluation, offering many benefits, making it a preferred and mandatory item [56,57]. The MTT assay is a fast and colorimetric method for rapidly evaluating cell proliferation and cytotoxicity by measuring cell metabolism or function [58]. In this study, the cytotoxicity effect of DMARE, MeJA-DMARE, and 3,5-DCQA on RAW 264.7 and A549 lung cancer cell lines was detected by the MTT assay. Our research found minimal toxicity in RAW cells when exposed to 3,5-DCQA within the concentration range of 31.25–500 μg/mL for 24 h, compared to the control. Concurrently, DMARE and MeJA-DMARE demonstrated low toxicity at concentrations below 250 μg/mL, as illustrated in (Figure 4A). Consequently, we selected 250 μg/mL for further experiments involving DMARE and MeJA-DMARE with RAW 264.7 cells. However, it is important to note that MeJA-DMARE and 3,5-DCQA exhibited elevated toxicity levels in A549 lung cancer cells when administered at concentrations exceeding 250 μg/mL, resulting in a 25% reduction in cell proliferation (Figure 4B). Similarly, for experiments conducted on A549 cell lines, we opted to use 250 μg/mL for DMARE and MeJA-DMARE based on these findings.

### 3.4. The Enhanced Anti-Inflammatory Activities 

#### 3.4.1. Inhibited NO and ROS Production 

Nitric oxide (NO) is synthesized by inducible NO synthase (iNOS) in activated macrophages and represents a pivotal inflammatory mediator. Physiologically, it triggers diverse detrimental reactions, encompassing tissue damage, septic shock, and apoptosis. The iNOS gene expression is prompted by exposure to microbial components like LPS in various inflammatory and tissue cells [59,60]. Conversely, excessive NO production in macrophages has the potential to disrupt and impair both normal and adjacent cells [61]. Hence, a drug that inhibits NO formation is acknowledged as a therapeutic remedy for addressing inflammation and managing cancer. 

We investigated the anti-inflammatory potential of DMARE, MeJA-DMARE, and 3,5-DCQA on Raw 264.7 cells. The cells were treated with DMARE, MeJA-DMARE, and 3,5-DCQA, followed by exposure to LPS at 1 μg/mL for 24 h. For comparison, our study included the usage of a well-known nitric oxide inhibitor, L-NMMA, as a positive control. As depicted in (Figure 5A), there is a notable increase in NO production in the LPS-treated cells compared to the control group. Conversely, NO production demonstrated a dose-dependent reduction in DMARE, MeJA-DMARE, and 3,5-DCQA-treated LPS-induced cells. DMARE and MeJA-DMARE exhibited substantial and statistically significant decreases in NO production, even at the lowest tested concentration (62.5 μg/mL), with reductions of 41.4% and 40.5%, respectively, compared to LPS-treated cells. In contrast, our reference compound, 3,5-DCQA, demonstrated significant reductions in NO production when compared to cells treated with LPS. Within each treatment group, the extracts from MeJA-DMARE displayed a superior ability to inhibit NO production compared to the extraction from untreated roots. This demonstrates the higher potential of the extracts from MeJA-DMARE on anti-inflammatory activities.

In contrast, LPS triggers the generation of reactive oxygen species (ROS) by activating NADPH-oxidase in macrophages. ROS act as mediators of cellular injury, contributing to the initiation of cellular damage during endotoxemia. Additionally, ROS are believed to regulate inflammatory gene expression through the redox-based activation of the *NF-κB* signaling pathway [62]. Reactive oxygen species (ROS) play crucial roles in oxidative stress, contributing to various pathophysiological processes, including chronic inflammation and autoimmune diseases. In the context of bacterial inflammation, excessive ROS can not only destroy bacteria but also harm human cells through oxidative stress. Additionally, pro-inflammatory cytokines can induce excessive ROS generation, leading to tissue inflammation and damage [63]. To assess the suppressive impact of DMARE, MeJA-DMARE, and 3,5-DCQA on LPS-induced ROS generation, RAW 264.7 cells were exposed to different preparations of these materials, either with or without LPS, for 24 h. As illustrated in (Figure 5B), the production of ROS was notably higher in the group exposed to LPS. Conversely, when LPS-induced cells were treated with DMARE, MeJA-DMARE, and 3,5-DCQA, there was a dose-dependent decrease in ROS production. Significantly decreased ROS generation was observed at MeJA-DMARE concentrations exceeding 125 μg/mL. MeJA-DMARE exhibited a greater capacity to suppress ROS production than DMARE in LPS-activated RAW 264.7 cells in each treatment group. These results indicated that our DMARE can significantly decrease ROS generation in LPS-RAW cells, thus protecting RAW cells from inflammation and damage. This underscores the elevated potential of MeJA-DMARE for anti-inflammatory purposes. *D*. *morbifera* harbors a range of biologically active compounds, such as flavonoids and phenolics, known for their antioxidant and anticancer properties. Furthermore, extracts derived from *D. morbifera* have demonstrated anti-inflammatory effects by inhibiting the LPS-induced inducible nitric oxide synthase (iNOS) in RAW 264.7 macrophages [64]. Furthermore, the MeJA-treated *D. morbifera* adventitious roots exhibited an elevated content of another phenolic compound, 3,5-DCQA. This increase contributed to enhanced anti-inflammatory activity, as indicated by the reduced production of NO and ROS in LPS-induced RAW cells, consistent with previous reports [65,66].

#### 3.4.2. The Inhibition of the Increased Levels of Inflammation-Related Cytokines

Inflammation acts as a defense mechanism against harmful stimuli. It is vital to terminate the inflammatory response to prevent tissue damage efficiently. Failure to do so can lead to chronic inflammation and cellular damage [67]. Among the intracellular signaling pathways involved in cytokine production, transcription factor NF-κB and mitogen-activated protein kinase (MAPK) cascades have been identified to play crucial roles [68]. NF-κB is implicated in the pathogenesis of inflammatory diseases and holds significant regulatory control over the transcription of pro-inflammatory mediators such as *iNOS*, *COX-2*, *TNF-α*, *IL-1β*, and *IL-6* [69]. Therefore, we investigated the impact of MeJA-DMARE on the expression of these factors using qRT-PCR. As depicted in Figure 5C–H, in the group subjected to LPS treatment, there was a significant upregulation in the mRNA expression of *COX-2*, *TNF-α*, *iNOS*, *IL-6*, and *IL-1β* (8.86, 7.38, 11.67, 8.70, and 5.04-fold as compared to untreated cells, respectively). Nevertheless, the MeJA-DMARE attenuated gene expression at 250 μg/mL by 1.79, 3.08, 3.14, 1.13, and 2.14-fold, respectively, compared to untreated cells. Our findings conclusively demonstrate that MeJA-DMARE exhibits potent anti-inflammatory properties.

### 3.5. Anti-Lung Cancer Activities

#### 3.5.1. Increased ROS Production

Reactive oxygen species (ROS) oversee critical cellular functions, including cell cycle regulation and apoptosis [70,71]. To illustrate, heightened ROS production within cancer cells triggers signaling pathways essential for tumor initiation, promotion, and advancement while also contributing to the tumor’s resistance to chemotherapy. Nonetheless, the highly reactive hydroxyl radical generated from hydrogen peroxide can cause extensive oxidation of proteins, lipids, and DNA, resulting in substantial damage or genomic instability [72,73]. Cancer cells generally display higher ROS production levels than normal cells [74]. However, cancer cells also exhibit enhanced activity of antioxidant enzymes to counterbalance these elevated ROS levels. Therefore, therapeutic approaches that either boost ROS production or reduce antioxidant defenses in cancer cells may trigger multiple cell death pathways, thereby restraining cancer progression [71,74,75]. 

Considering the crucial influence of ROS on cell proliferation and survival, we investigated whether MeJA-treated *D. morbifera* adventitious roots could stimulate ROS production in human lung cancer cells. For this purpose, we assessed cellular and mitochondrial ROS levels by employing DCFH-DA. DCFH is impermeable to cell membranes and undergoes rapid oxidation to form highly fluorescent 2,7-dichlorofluorescein in the presence of intracellular ROS [76]. The fluorescence intensity is indicative of the oxidative stress level. In contrast to the control group, a dose-dependent increase in intracellular ROS production was observed in the malignant A549 cells following treatment with MeJA-DMARE and 3,5-DCQA at concentrations of 250 µg/mL (Figure 6A). Bioactive compounds can trigger apoptosis in cancer cells by promoting the accumulation of ROS [77]. Mildly elevated levels of ROS have been demonstrated to promote tumor growth by interacting with various proteins and intracellular signaling pathways involved in tumor proliferation and survival. However, high levels of ROS can be fatal to cancer cells by triggering senescence and inducing multiple forms of cell death, including apoptosis, autophagy, and ferroptosis [78]. Our findings revealed that MeJA-treated *D. morbifera* adventitious roots can promote ROS generation under the influence, thus inhibiting tumor growth. The generation of intracellular ROS in A549 cells due to the impact of 3,5-DCQA was definitively confirmed [79]. Previous reports have highlighted ROS as a primary target for inhibiting cancer cells, and our conducted experiment with 3,5-DCQA induced robust oxidative stress responses in A549 cells [80].

#### 3.5.2. Wound Healing

Collective cell migration is fundamental to processes, such as wound repair, cancer invasion and metastasis, immune responses, angiogenesis, and embryonic morphogenesis. The wound-healing process is intricate, involving cellular and biochemical mechanisms to restore damaged tissue structure. It encompasses dynamic interactions and communication among various cell types, interactions with extracellular matrix molecules, and the controlled production of soluble mediators and cytokines. In the context of cutaneous wound healing, skin cells migrate from the wound edges into the wound site to facilitate the restoration of skin integrity. Assessing cell migration in vitro is a valuable assay to quantitatively measure changes in cell migratory capability in response to experimental interventions [81]. 

Identifying targets to inhibit the migration and invasion of cancer cells into other tissues is challenging in preventing and treating metastatic cancer. We employed a scratch migration assay to assess the effects of DMARE, MeJA-DMARE, and 3,5-DCQA on A549 cell migration. Additionally, we conducted a wound closure test to compare the migration (%) of A549 lung cancer cells before and after treatment with DMARE, MeJA-DMARE, and 3,5-DCQA (at doses of 250 µg/mL). Following treatment with DMARE, MeJA-DMARE, and 3,5-DCQA, we observed a reduction in migratory cells, implying that these compounds inhibit the lateral movement of A549 cells. Moreover, as shown in Figure 6B,C, MeJA-DMARE exhibited a more potent inhibitory effect than DMARE. Consequently, our study suggests that MeJA-DMARE inhibits A549 cell proliferation and in vitro metastasis. The capacity of MeJA-DMARE to hinder growth may indicate their potential as treatments for lung cancer.

#### 3.5.3. Apoptotic Gene Expression in Lung Cancer Cells

The cellular pathway involving Kelch-like ECH-associated protein 1 (Keap1) and nuclear factor erythroid 2-related factor 2 (Nrf2) is a significant mechanism that protects normal cells from oxidative and xenobiotic damage [82]. Activating Nrf2, a critical transcription factor in countering oxidative stress, is a potent strategy for preventing cancer from exposure to environmental carcinogens [83]. The MAPK pathway is a prevalent signaling route responsible for coordinating how cells react to various external signals. There are two primary MAP kinase cascades: p38 MAPK and JNK. All of these kinases probably play a role in the initial pathways that activate Nrf2 [84]. As depicted in Figure 7, the mRNA expression levels of *p 38 MAPK* and *JNK* detected by qRT-PCR were lowest in the control group, and MeJA-DMARE could significantly upregulate these expressions in A549 lung cancer cells (*p 38 MAPK* and *JNK* increased 3.90 and 3.35-fold); additionally, the expression levels of *Nrf2*, *HO-1*, and *CAT* were highest in the MeJA-DMARE-treated group, and MeJA-DMARE could significantly downregulate these expressions in A549 lung cancer cells (*Nrf2*, *HO-1*, and *CAT* decreased by 0.42-fold, 0.94-fold, and 0.47-fold).

In the mitochondrial pathway, apoptosis relies on the release of cytochrome c from mitochondria into the cytosol, a process initiated by the interaction between mitochondria and one or more proteins from the Bcl-2 family. Therefore, several proteins within the Bcl-2 family are acknowledged as key regulators of the apoptotic process, playing a critical role as a checkpoint in apoptotic pathways, positioned upstream of irreversible damage to cellular components [85,86]. As shown in Figure 8, MeJA-DMARE significantly downregulated the expression of *Bcl-2* by 0.31-fold and significantly upregulated the mRNA expression of *Caspase 3/9* and *Bax* by 2.05, 3.17 and 5.28-fold, respectively. Consequently, this study revealed that MeJA-DMARE effectively inhibited the expression of apoptosis-related genes in lung cancer cells.

## 4. Conclusions

Phytochemicals are crucial antioxidants in reducing oxidative stress, protecting cells from damage, and potentially preventing cancer and various diseases. This study focused on cultivating adventitious roots in *Dendropanax morbifera* and identifying compounds in DMAR. The results showed that DMAR had similar levels of total phenolic and total flavonoid content and displayed close antioxidant properties with DMLE. The study explored the impact of MeJA treatment on DMAR’s biomass and the accumulation of secondary metabolites, particularly 3,5-DCQA. The most effective concentration for 3,5-DCQA production was 40 µM for 36 h. MeJA-DMARE exhibited strong anti-inflammatory properties by reducing the NO and ROS production in LPS-induced RAW 264.7 cells. It also downregulated the mRNA expression of inflammation-related cytokines. Additionally, MeJA-DMARE demonstrated anti-lung cancer activity by increasing ROS production in A549 lung cancer cells and inhibiting their migration at a 250 µg/mL concentration. It was also involved in regulating the apoptosis of lung cancer cells through the Bcl-2 and p38 MAPK pathways. In conclusion, MeJA-DMARE shows promise as a sustainable and novel material for potential pharmaceutical applications due to its antioxidant, anti-inflammatory, and anticancer properties.

## Figures and Tables

**Figure 1 biomolecules-14-00705-f001:**
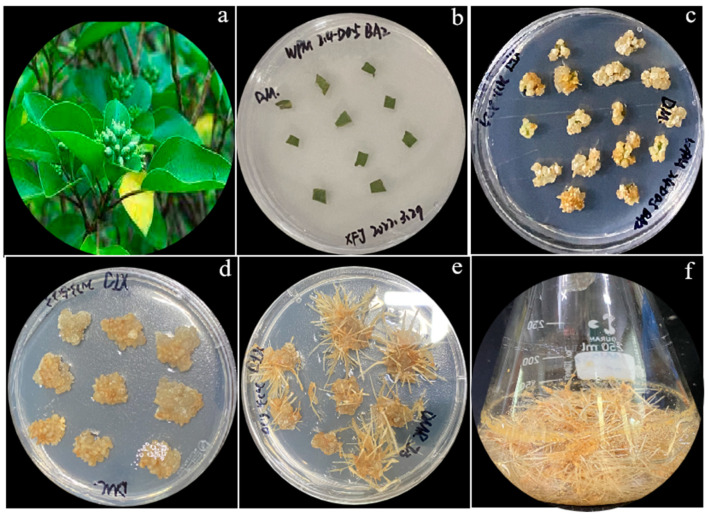
In vitro propagation of DMAR. (**a**), natural plant; (**b**), leaf explant—1 day’s culture; (**c**), callus—4 weeks’ culture; (**d**), callus—6 weeks’ culture; (**e**), adventitious root—2.5 months’ culture; and (**f**), adventitious root—4 months’ culture.

**Figure 2 biomolecules-14-00705-f002:**
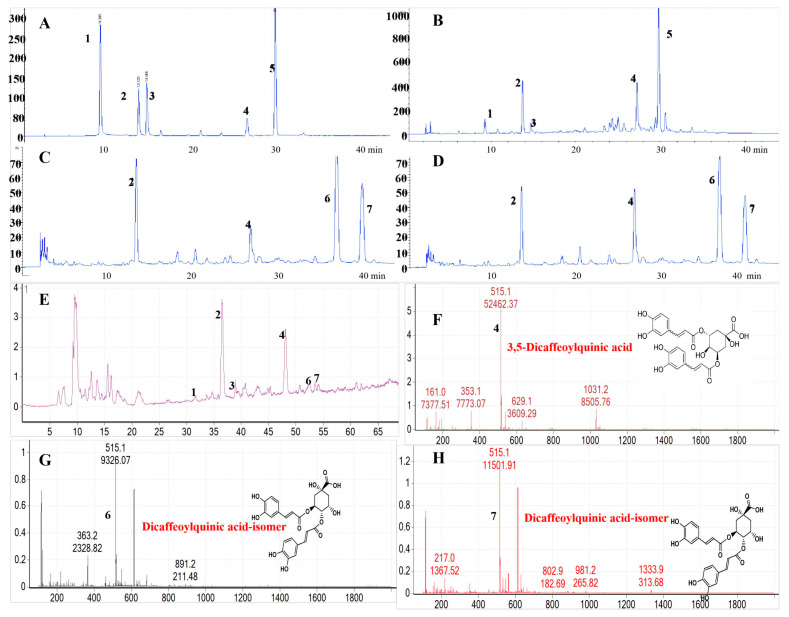
Compounds analysis by using HPLC and LC-MS. (**A**) Five standards: **1**, Neo-chlorogenic acid; **2**, Chlorogenic acid; **3**, Cryptochlorogenic acid; **4**, 3,5-Dicaffeoylquinic acid (3,5-DCQA); and **5**, Rutin; (**B**) compounds in DMLE; (**C**) compounds in DMARE; (**D**) compounds in MeJA-DMARE; (**E**) LC-MS analysis of MeJA-DMARE; (**F**) LC-MS analysis shows that the peak of the increased compound **4** in MeJA-DMARE is 3,5-DCQA at a retention time between 47.672 and 48.252 min; (**G**) LC-MS analysis shows that the peak of compound **6** in MeJA-DMARE is Di-caffeoylquinic acid isomer at a retention time between 52.180 and 53.151 min; and (**H**) LC-MS analysis shows that the peak of compound **7** in DMARE is Di-caffeoylquinic acid isomer at a retention time between 54.340 and 55.185 min.

**Figure 3 biomolecules-14-00705-f003:**
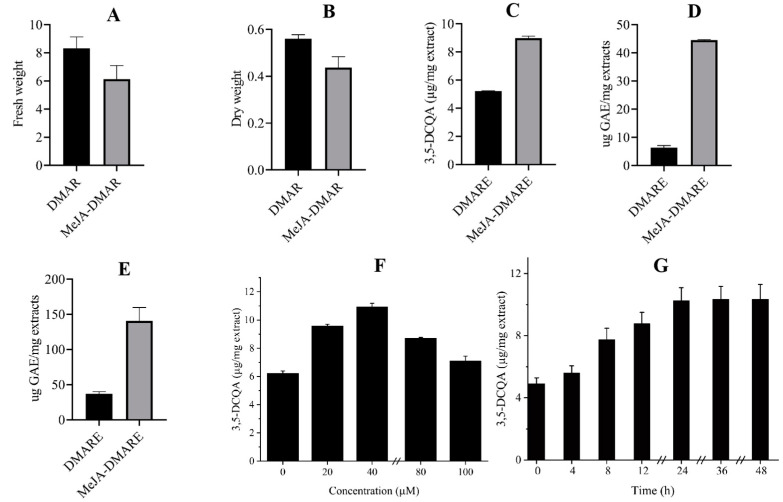
The effect of MeJA on the growth and biomass of DMAR. (**A**) The effect on fresh weight; (**B**) the effect on dry weight; (**C**) the effect on the 3,5-DCQA accumulation; (**D**) the antioxidant activity comparison analyzed by DPPH; (**E**) the antioxidant activity comparison analyzed by reducing power; (**F**) the influence of concentration on the 3,5-DCQA accumulation; and (**G**) the impact of exposure time on the 3,5-DCQA accumulation.

**Figure 4 biomolecules-14-00705-f004:**
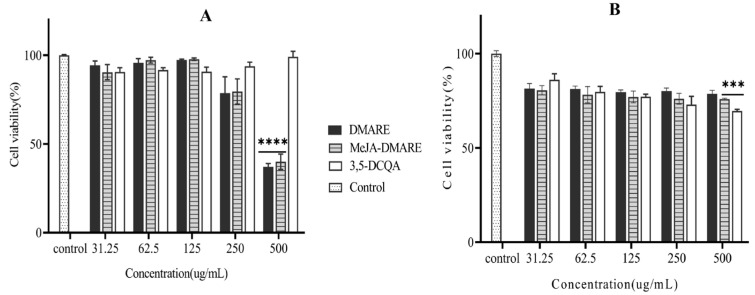
Cytotoxicity of DMARE, MeJA-DMARE, and 3,5-DCQA on (**A**) RAW 264.7 and (**B**) A549 lung cancer cells. The graph displays the mean ± SD values from three replicate measurements. Statistically significant differences compared to the control group were mentioned by *** *p* < 0.001 and **** *p* < 0.0001.

**Figure 5 biomolecules-14-00705-f005:**
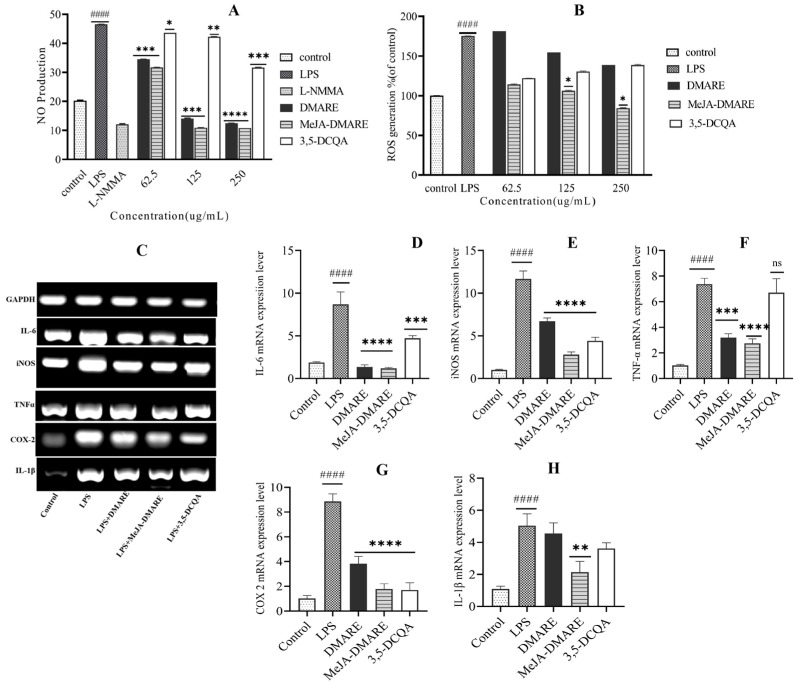
The anti-inflammatory activities of DMARE, MeJA-DMARE, and 3,5-DCQA on RAW 264.7 cells. (**A**) NO generation; (**B**) ROS production; (**C**) the mRNA expression of five selected genes is depicted as bands in agarose gel electrophoresis; and (**D**–**H**) the gene expression levels were quantified by qRT-PCR. From (**D**-**H**), the genes are *IL-6*, *iNOS*, *TNF-α*, *COX-2*, and *IL-1β*, respectively. The graph displays the mean ± SD values from three replicate measurements. Statistically significant differences compared to the control group are denoted by #### *p* compared to the control group; * *p* < 0.05; ** *p* < 0.01; *** *p* < 0.001; and **** *p* < 0.0001.

**Figure 6 biomolecules-14-00705-f006:**
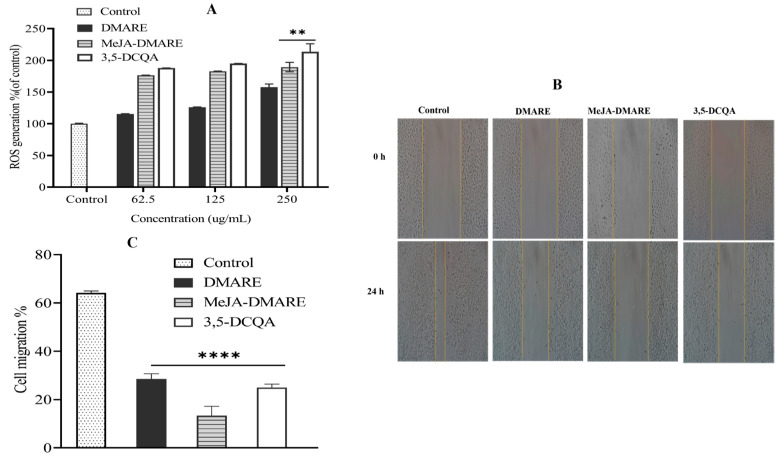
The anti-lung cancer activities of DMARE, MeJA-DMARE, and 3,5-DCQA on A 549 cells. (**A**) The capacity of MeJA-DMARE to promote intracellular ROS production; (**B**) the migration of lung cancer cells. Image J software 18.0 version was employed to assess the cell-free area in the scratched region; (**C**) the ratio of cell migration following a 24-hour treatment compared to the control. The graph displays the mean ± SD values from three replicate measurements. Statistically significant differences compared to the LPS-induced group are denoted by ** *p* < 0.01 and **** *p* < 0.0001.

**Figure 7 biomolecules-14-00705-f007:**
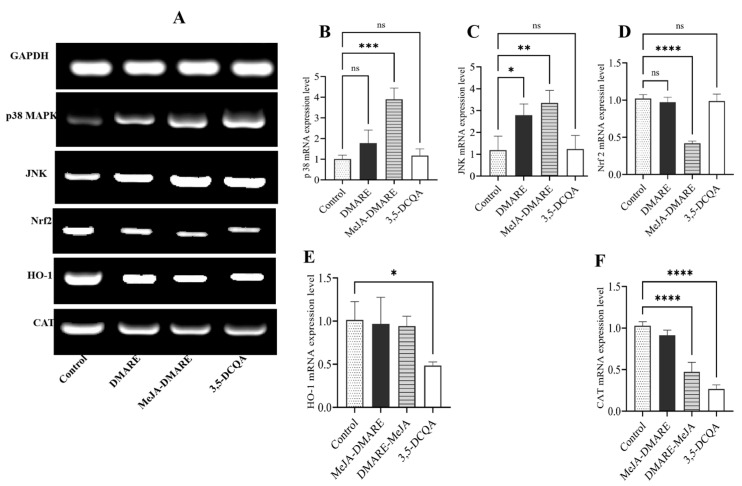
The mRNA expression levels of apoptosis-related genes on A549 cells in MAPK pathway. (**A**) The mRNA expression levels of five selected genes are visualized and presented as agarose gel electrophoresis bands. (**B**–**F**) The gene expression levels were quantified using qRT-PCR. From (**B**–**F**), the genes are *p38 MAPK*, *JNK*, *Nrf-2*, *HO-1*, and *CAT*, respectively. Statistically significant differences compared to the control group are denoted by * *p* < 0.05; ** *p* < 0.01; *** *p* < 0.001; and **** *p* < 0.0001, and ns means no significant difference. All treatments were performed in triplicate.

**Figure 8 biomolecules-14-00705-f008:**
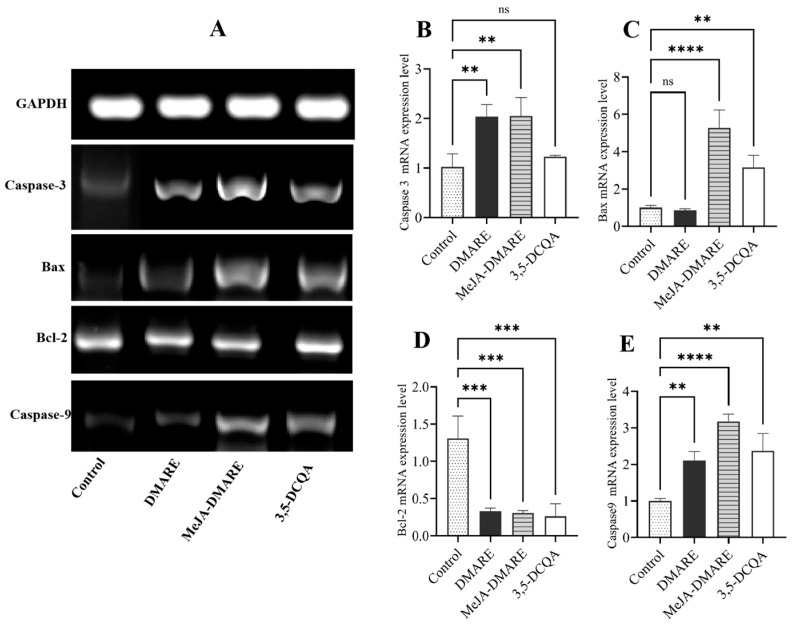
The mRNA expression levels of apoptosis-related genes on A549 cells in Bcl-2 pathway. (**A**) The mRNA expression levels of five selected genes are visualized and presented as agarose gel electrophoresis bands. (**B**–**E**) The gene expression levels were quantified by qRT-PCR. From (**B**–**E**), the genes are *Caspase-3*, *Bax*, *Bcl-2*, *HO-1*, and *Caspase-9*, respectively. Statistically significant differences compared to the control group are denoted by * *p* < 0.05;** *p* < 0.01; *** *p* < 0.001; and **** *p* < 0.0001, and ns means no significant difference. All treatments were performed in triplicate.

**Table 1 biomolecules-14-00705-t001:** Comparison of TPC, TFC, and antioxidant activities of DMARE and DMLE.

Samples	TPC	TFC	Antioxidant Activities	
			DPPH	Reducing Power
	(μg GAE/mg Extracts *)	(μg RE/mg Extracts **)	(μg GAE/mg Extracts)	(μg GAE/mg Extracts)
DMARE	116.5 ± 1.4 ^b^	105.8 ± 12.4	8.7 ± 0.1	311.5 ± 26.3
DMLE	140.2 ± 1.9 ^a^	98.3 ± 2.6	8.5 ± 0.4	315.7 ± 7.0

μg GAE/mg extracts *: μg Gallic acid equivalents (μg GAE)/mg dry weight of DMARE and DMLE; μg RE/mg extracts **: μg Rutin equivalents (μg GAE)/mg dry weight of DMARE and DMLE. The results are shown as means ± standard deviation (3 replicates). Values in the same column separated by different letters (a, b) differ significantly. *p* < 0.05 was considered statistically significant.

## Data Availability

The data are contained within the article.

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
