# Peer review of "3,5-DCQA as a Major Molecule in MeJA-Treated Dendropanax morbifera Adventitious Root to Promote Anti-Lung Cancer and Anti-Inflammatory Activities"

_biomolecules, 2024, doi:10.3390/biom14060705_

Round 1

Reviewer 1 Report

Comments and Suggestions for Authors

In their manuscript Hu et al. explored the use of methyl jasmonate (MeJA) in the in vitro cultivation of D. morbifera adventitious roots (DMAR) and evaluated its impact on secondary metabolite production in DMAR, optimizing concentration and exposure time. Further, they evaluated the anti-inflammatory and anti-lung cancer activities, and looked for the changes in the expression of related genes, detected by qRT-PCR. The authors concluded that  MeJA-treated DMARE with increased 3,5-DCQA production holds significant promise as a sustainable and novel material for pharmaceutical applications.

The manuscript and study design are appropriate for answering the research questions and test the hypothesis, but not clearly presented, and a few observations need to be taken into account, as outlined below. The manuscript’s results should be reproducible based on the details given in the methods section, but some details and appropriate controls are missing, and the paper contains several serious mistakes.

1) The culture media usually used for the described cell lines should contain L-glutamine.

2) Table 1: please change ug in mg. Values of TFC for DMARE and DMLE do not seem to be very different since +SD values are taken into account.

3) The cytotoxicity profiles of the compounds were performed on RAV 264.7 and A549 cell lines using various concentrations (0-500  μg/ml). In the manuscript authors said that solutions for HPLC analysis were prepared by solving the powder extracts in distilled water and filtered through 0.45 mm filters. How were prepared the sterile stock solutions for the cytotoxicity assays, since 0.45 mm filtering  do not sterilize them, so they cannot be used in cell culture treatments? Filtering through a 0.22 mm filter could retain a large part of the plant extract on it.

4) The cytotoxicity assays might be influenced by the potential interference of different chemical compounds or plant extracts with MTT dye. Therefore, the authors should perform an interference test of all the compounds under study with the viability dye in the absence of cells. If any interference occurs for a compound, values should be subtracted. The formula used for calculating the percentages of cell viabilities after performing the cytotoxicity assays should also be inserted in the manuscript.

5) Figure 4B shows that the reduction in cell proliferation is 25-30% for 250 mg/ml treatments, and not 75%, as written. The percentages of cell viability were reduced to 75-80%, which is not equal to reduction in cell proliferation.

6) Lines 235-236: ”Different concentrations of DMARE, MeJA-235 DMARE and 3,5-DCQA were pretreated into RAW 264.7 cells for 1 h”. The correct statement should be: ”RAW 264.7 cells were pretreated for 1 h with different concentrations of DMARE, MeJA-235 DMARE and 3,5-DCQA”.

7) The correct name for Nrf2 is nuclear factor erythroid 2-related factor 2.

8) Please correct the statement from lines 562-563:: ”There are three primary MAP kinase cascades: p38 MAPK and JNK”.

Comments on the Quality of English Language

Moderate editing of English language required

Author Response

Response to Reviewer 1 Comments

Comments and Suggestions for Authors

In their manuscript Xu et al. explored the use of methyl jasmonate (MeJA) in the in vitro cultivation of D. morbifera adventitious roots (DMAR) and evaluated its impact on secondary metabolite production in DMAR, optimizing concentration and exposure time. Further, they evaluated the anti-inflammatory and anti-lung cancer activities, and looked for the changes in the expression of related genes, detected by qRT-PCR. The authors concluded that  MeJA-treated DMARE with increased 3,5-DCQA production holds significant promise as a sustainable and novel material for pharmaceutical applications.

The manuscript and study design are appropriate for answering the research questions and test the hypothesis, but not clearly presented, and a few observations need to be taken into account, as outlined below. The manuscript’s results should be reproducible based on the details given in the methods section, but some details and appropriate controls are missing, and the paper contains several serious mistakes.

[Authors’ response] Thank you very much for taking the time to evaluate our manuscript (biomolecules-3035669) and your positive comments and excellent suggestions. We tried our best to improve our manuscript by reflecting your comments in this revised manuscript. We hope this revision meets your expectations. Thanks again and have a great day!

point 1: The culture media usually used for the described cell lines should contain L-glutamine.

Response 1: Thank you very much for your feedback. We are very sorry for the missed information. In this revised manuscript, we added this important information, as shown in lines 131 and 235, marked in red.

point 2: Table 1: please change ug in mg. Values of TFC for DMARE and DMLE do not seem to be very different since +SD values are taken into account.

Response 2: Thank you very much for your positive comments. We calculated the total phenolics content and total flavonoide content in this work. The TPC/TFC results were given as μg of gallic acid /rutin equivalent (RE) per mg of extract. We removed the significant values of TFC, DPPH and reducing power results for DMARE and DMLE since they are not very different, as shown in Table 1, in this revised manuscript.

point 3: The cytotoxicity profiles of the compounds were performed on RAV 264.7 and A549 cell lines using various concentrations (0-500  μg/ml). In the manuscript authors said that solutions for HPLC analysis were prepared by solving the powder extracts in distilled water and filtered through 0.45 mm filters. How were prepared the sterile stock solutions for the cytotoxicity assays, since 0.45 mm filtering  do not sterilize them, so they cannot be used in cell culture treatments? Filtering through a 0.22 mm filter could retain a large part of the plant extract on it.

Response 3: Thank you very much for your positive comments.  To address this, we need clarification on how the sterile stock solutions were prepared for the cytotoxicity assays. Typically, sterile solutions for cell culture can be achieved by using 0.22 μm filters.

In this work, all the sample extracts and 3,5-DCQA standard were filtered by 0.22 μm filters before treating cells. Only the amount of 3,5-DCQA, was increased after the treatment of methyl jasmonate. So the target of this work was to answer whether this increased compound could enhance the anti inflammation and anti-lung cancer activity. As indicated in our results, the bioactivities were also enhanced with the increasement of 3,5-DCQA. Meanwhile, we evaluated the biological activities of 3,5-DCQA in both cell lines, and found it effectively in inhibited inflammation and lung cancer growth, indicating that this compound can pass this filter. Previous reports also highlighted the anti-inflammation and anti-lung cancer activities of 3,5-DCQA, as the references 66, 67, 80 and 81 that we cited.

point 4: The cytotoxicity assays might be influenced by the potential interference of different chemical compounds or plant extracts with MTT dye. Therefore, the authors should perform an interference test of all the compounds under study with the viability dye in the absence of cells. If any interference occurs for a compound, values should be subtracted. The formula used for calculating the percentages of cell viabilities after performing the cytotoxicity assays should also be inserted in the manuscript.

Response 4: Thank you very much for your useful comments. In the MTT experiment, we remove the culture medium containing the plant extract and then use DMSO to dissolve the formazan crystals to measure the absorbance. This ensures that the crude extracts do not interfere with the experimental results. The color development is due to the reaction with the formazan crystals within the cells, not with the culture medium or crude extracts. We are sorry that we didn’t clearly clarify the protocol, we rewrote this protocol in this revised manuscript, as shown in lines 248-253. And the formula used for calculating the percentages of cell viabilities after performing the cytotoxicity assays was inserted in the revised manuscript also, as shown in line 251.

point 5: Figure 4B shows that the reduction in cell proliferation is 25-30% for 250 mg/ml treatments, and not 75%, as written. The percentages of cell viability were reduced to 75-80%, which is not equal to reduction in cell proliferation.

Response 5: Thank you very much for your valuable comments. We are sorry for the mistake we made while writting. We corrected it  in this revised manuscript, as shown in line 450.

point 6: Lines 235-236: ”Different concentrations of DMARE, MeJA-235 DMARE and 3,5-DCQA were pretreated into RAW 264.7 cells for 1 h”. The correct statement should be: ”RAW 264.7 cells were pretreated for 1 h with different concentrations of DMARE, MeJA-235 DMARE and 3,5-DCQA”.

Response 6: Thank you very much for your valuable comments. We are sorry for the improper writting. We corrected it in this revised manuscript, as shown in lines 500-502.

point 7: The correct name for Nrf2 is nuclear factor erythroid 2-related factor 2.

Response 7: Thank you very much for your valuable comments. We are sorry for the mistake we made while writting. We corrected it  in this revised manuscript, as shown in lines 600-601.

point 8: Please correct the statement from lines 562-563:: ”There are three primary MAP kinase cascades: p38 MAPK and JNK”.

Response 8: Thank you very much for your valuable comments. We are sorry for the mistakes. We corrected it in this revised manuscript, as shown in line 606.

point 9: Comments on the Quality of English Language: Moderate editing of English language required

Response 9: We tried our best to improve our English expression in this revised manuscript according to your valuable comments. 

Reviewer 2 Report

Comments and Suggestions for Authors

In my opinion, this is a very good, interesting and quality work.

This work will undoubtedly be of interest to the readers of the Biomolecules journal.

However, it is recommended to make the following corrections and additions before publication:

Main questions:

1. Based on the data obtained, two opposite effects are observed: 1) on the one hand a decrease in the level of ROS in RAW 264.7 cells, which is associated with anti-inflammatory activity, 2) and on the other hand, on the contrary, an increase in the level of ROS in A549 lung cancer cells, which determines anticancer activity. This begs the question: does the increase or decrease in ROS levels under the influence of such biologically active compounds depend on each specific cell culture? I would recommend clarifying this point and describing an explanation of these two opposing effects in the discussion of the results and in the conclusion.

2. Starting with the introduction, the authors explain the connection between inflammation and the occurrence of lung cancer and, accordingly, further in their experiments they study anti-cancer activity on lung cancer cells. But the question remains, what is the potential of the phytochemicals being studied against other forms of cancer? Have additional studies been carried out on other cancer cell lines? Or maybe there is literature data on this issue?

3. In my opinion the introduction section lacks any specific examples of chemotherapy currently used. Anticancer drugs (nucleoside and non-nucleoside) used in medicinal practice. Their advantages and disadvantages.

4. It is not clear where 3,5-DCQA was obtained in its pure form? Isolated from an extract mixture through the experiment? Purchased or synthesized? This moment was not described in the experiment. If it is isolated from a mixture, then proof of purity and structure is needed.

Minor notes:

1. Table 1 contains footnotes a, b, *, **, but there is no explanation of the footnotes under the table. 2. Figure 4. The graph has asterisk footnotes, but no explanation. Same for all graphs in Figures 5, 6 and 7. Also Figure 5 is fuzzy. D,E,F overlap each other.

3. There is a mistake in Section 2.7 name. Need “TFC” instead of “TPC”.

There may be other minor typos that I missed, so I want to advise the authors to carefully review the entire text of the article and correct minor typos.

Author Response

Response to Reviewer 2 Comments

In my opinion, this is a very good, interesting and quality work.

This work will undoubtedly be of interest to the readers of the Biomolecules journal.

However, it is recommended to make the following corrections and additions before publication:

[Authors’ response] Thank you very much for evaluating our manuscript and for your positive comments and excellent suggestions. We tried our best to improve our manuscript by reflecting on your comments in this revised manuscript. We hope this revision meets your expectations. Thanks again and have a nice day!

point 1: Based on the data obtained, two opposite effects are observed: 1) on the one hand a decrease in the level of ROS in RAW 264.7 cells, which is associated with anti-inflammatory activity, 2) and on the other hand, on the contrary, an increase in the level of ROS in A549 lung cancer cells, which determines anticancer activity. This begs the question: does the increase or decrease in ROS levels under the influence of such biologically active compounds depend on each specific cell culture? I would recommend clarifying this point and describing an explanation of these two opposing effects in the discussion of the results and in the conclusion.

Response 1: Thank you for your feedback. We appreciate your suggestion to clarify the relationship between ROS levels and the specific cell cultures used. Based on the data obtained, two contrasting effects are observed: a decrease in the level of ROS in RAW 264.7 cells, indicative of anti-inflammatory activity, and an increase in the level of ROS in A549 lung cancer cells, suggestive of anticancer activity. It would be beneficial to further investigate this point and provide an explanation for these two opposing effects. We  included a detailed discussion to address these observations in the manuscript. In this revised manuscript, we provided more explanations, as shown in lines 487-492 and 559-563 marked in red.

point 2: Starting with the introduction, the authors explain the connection between inflammation and the occurrence of lung cancer and, accordingly, further in their experiments they study anti-cancer activity on lung cancer cells. But the question remains, what is the potential of the phytochemicals being studied against other forms of cancer? Have additional studies been carried out on other cancer cell lines? Or maybe there is literature data on this issue?

Response 2: Thank you for your insightful comments. To address these points, we reviewed existing literature to explore the effects of these phytochemicals on other cancer types and discuss any relevant studies that have investigated their broader anticancer potential, and we cited these references in this revised manuscript, as shown in lines 80-83. Additionally, future experiments will aim to test these compounds on a variety of cancer cell lines to comprehensively evaluate their efficacy across different cancer forms.

point 3: In my opinion the introduction section lacks any specific examples of chemotherapy currently used. Anticancer drugs (nucleoside and non-nucleoside) used in medicinal practice. Their advantages and disadvantages.

Response 3: Thank you very much for your valuable comments. We added more related information about the examples of chemotherapy currently used, and advantages and disadvantages of anticancer drugs (nucleoside and non-nucleoside) used in medicinal practice in this revised manuscript, as shown in lines 53-65.

point 4: It is not clear where 3,5-DCQA was obtained in its pure form? Isolated from an extract mixture through the experiment? Purchased or synthesized? This moment was not described in the experiment. If it is isolated from a mixture, then proof of purity and structure is needed.

Response 4: Thank you for your feedback. We are sorry that we missed the related information about compound 3,5-DCQA. We added it in this revised manuscript, as shown in lines 129-130.

Minor notes:

  1. Table 1 contains footnotes a, b, *, **, but there is no explanation of the footnotes under the table.

Response: Thank you for your feedback. We added the explanations of these footnotes in this evised manuscript, as shown in lines 333-336.

  1. Figure 4. The graph has asterisk footnotes, but no explanation. Same for all graphs in Figures 5, 6 and 7. Also Figure 5 is fuzzy. D,E,F overlap each other.

Response: Thank you for your feedback. We added the explanations of these footnotes in this evised manuscript, as shown in lines 455-457; 532-535; 573-575; 618-620 and 635-637. And Figure 5 was reconstructed and put in this revised manuscript.

  1. There is a mistake in Section 2.7 name. Need “TFC” instead of “TPC”.

Response: Thank you for your feedback. We corrected it in this evised manuscript, as shown in lines 204 and 211. And Figure 5 was remade and put.

Round 2

Reviewer 1 Report

Comments and Suggestions for Authors

The authors answered to all my questions or comments, but I am not sure they have understoond the meaning of all of them. Therefore, I'll repeat some of my concerns:

point 1: "The culture media usually used for the described cell lines should contain L-glutamine". My question reffered to all cell culture media used in the present study, not only DMEM. Please insert the L-glutamine concentrations that were used, if you added it. Some of the commercial media contain stable L-glutamine, otherwhile you have to add it in all your cell culture media, including the one with RPMI-1640.

point 2: Table 1, please use the right symbol for micro.

point 4: "The cytotoxicity assays might be influenced by the potential interference of different chemical compounds or plant extracts with MTT dye. Therefore, the authors should perform an interference test of all the compounds under study with the viability dye in the absence of cells. If any interference occurs for a compound, values should be subtracted".

Even if the authors took off the media containing their vegetal extracts and replaced with MTT solution, the compounds that entered the cells might interfere with the colouring reagent. The MTT assay performed with all dilutions of the tested compounds in the absence of cells is absolutely necessary.

Comments on the Quality of English Language

Minor editing of English language required

Author Response

Response to Reviewer 1 Comments

Comments and Suggestions for Authors

The authors answered to all my questions or comments, but I am not sure they have understoond the meaning of all of them. Therefore, I'll repeat some of my concerns:

[Authors’ response] We apologize for any misunderstandings to your comments in previous responses. We appreciate your patience and would like to address your concerns in detail to ensure clarity and completeness. Thank you once again for your valuable insights and helping us improve our manuscript.

point 1: "The culture media usually used for the described cell lines should contain L-glutamine". My question referred to all cell culture media used in the present study, not only DMEM. Please insert the L-glutamine concentrations that were used, if you added it. Some of the commercial media contain stable L-glutamine, otherwise you have to add it in all your cell culture media, including the one with RPMI-1640.

Response 1: Thank you very much for your feedback. We apologize for not explaining this clearly in our Ms and previous response. All the media we used for cell culture already contain L-glutamine ( Dulbecco’s modified Eagle medium (DMEM) containing L-glutamine), RPMI 1640 culture medium (with L-glutamine) were purchased from Welgene Inc., (Gyeongsan-si, Republic of Korea). ), we did not add this substance artificially in our experiments. We revised it in manuscript (marked in red). Thank you again for pointing this out to us.

point 2: Table 1, please use the right symbol for micro.

Response 2: Thank you very much for your positive comments. We have corrected the symbol for micro.

point 4: "The cytotoxicity assays might be influenced by the potential interference of different chemical compounds or plant extracts with MTT dye. Therefore, the authors should perform an interference test of all the compounds under study with the viability dye in the absence of cells. If any interference occurs for a compound, values should be subtracted".

Even if the authors took off the media containing their vegetal extracts and replaced with MTT solution, the compounds that entered the cells might interfere with the colouring reagent. The MTT assay performed with all dilutions of the tested compounds in the absence of cells is absolutely necessary.

Response 4: We greatly appreciate your feedback and agree with your viewpoint. In our preliminary experiment, we tested for potential interference between the plant extract and the compounds mentioned in this study with MTT. The experimental results showed that none of the compounds reacted with MTT. We apologize deeply for not mentioning this in the manuscript and not fully addressing your concerns in our previous response.  

Thank you for your insightful comments regarding the potential interference of different chemical compounds or plant extracts with the MTT dye in our cytotoxicity assays. We will incorporate these considerations in future experiments to enhance the scientific rigor of our research. In revised manuscript, we provided a more detailed description of the experimental process (marked in red). 

Thank you again for your valuable comments.

Comments on the Quality of English Language: Minor editing of English language required

Response: Thank you for your feedback and for highlighting the need for minor edits to the English language in our manuscript. We carefully reviewed the manuscript and made the necessary corrections according to your suggestion. 

Thank you once again for your valuable suggestions.